# Delivery of microRNAs by Extracellular Vesicles in Viral Infections: Could the News be Packaged?

**DOI:** 10.3390/cells8060611

**Published:** 2019-06-18

**Authors:** Fabio Seiti Yamada Yoshikawa, Franciane Mouradian Emidio Teixeira, Maria Notomi Sato, Luanda Mara da Silva Oliveira

**Affiliations:** Laboratory of Dermatology and Immunodeficiencies, Department of Dermatology, Tropical Medical Institute, School of Medicine, University of São Paulo, 05403-000 São Paulo, Brazil; faseiti@gmail.com (F.S.Y.Y.); franciane.mteixeira@usp.br (F.M.E.T.)

**Keywords:** extracellular vesicles, miRNAs, RNA virus, DNA virus

## Abstract

Extracellular vesicles (EVs) are released by various cells and recently have attracted attention because they constitute a refined system of cell–cell communication. EVs deliver a diverse array of biomolecules including messenger RNAs (mRNAs), microRNAs (miRNAs), proteins and lipids, and they can be used as potential biomarkers in normal and pathological conditions. The cargo of EVs is a snapshot of the donor cell profile; thus, in viral infections, EVs produced by infected cells could be a central player in disease pathogenesis. In this context, miRNAs incorporated into EVs can affect the immune recognition of viruses and promote or restrict their replication in target cells. In this review, we provide an updated overview of the roles played by EV-delivered miRNAs in viral infections and discuss the potential consequences for the host response. The full understanding of the functions of EVs and miRNAs can turn into useful biomarkers for infection detection and monitoring and/or uncover potential therapeutic targets.

## 1. Introduction

Intercellular communication is essential for the homeostasis of biological systems. Among the many ways to share information, the ability of cells to release extracellular vesicles (EVs), which until a few decades ago were considered “cell dust”, has attracted much attention in scientific research recently. Their first description was in the 1980s, when vesicles with considerable size, released by the shedding of small areas of the plasma membrane of erythrocytes in culture, were observed under light microscopy [1].

EVs have received several names over time, including “shedding vesicles”, “microvesicles” and “ectosomes” [1] but, currently, we can categorize them into three different types: (i) exosomes, the term used for EVs ≤ 100 nm in diameter that originate from multivesicular bodies (MVBs); (ii) microvesicles, the term used for those whose diameter is 100–1000 nm, which are shed from the plasma membrane; and (iii) apoptotic bodies, the term for those with a diameter > 1000 nm, which are usually released by cells under apoptosis [2]. Nevertheless, it has been very difficult to distinguish each of these populations because they share similar markers, such as physical and biochemical characteristics, size, and density [3]. Therefore, we will use the general term EVs in order to study both exosomes and MVs in this text.

Different techniques have been described to isolate EVs and characterize their release, uptake, and cargo. The choice of the best method for EV isolation has been the object of great efforts in recent years, although techniques such as immunoblotting, fluorescent microscopy, and electron microscopy have all been used as standards to characterize and visualize EVs. In recent years, more fine-tuned techniques have emerged [4].

EVs are involved in a variety of biological and disease functions. EVs derived from dendritic cells (DCs) can act in antigen presentation, playing a crucial role in carrying and presenting functional MHC–peptide complexes to modulate antigen-specific CD8^+^ and CD4^+^ responses [5]. Platelet-derived EVs constitute the majority of circulating EVs and are preferentially associated with granulocytes and monocytes, while they scarcely interact with lymphocytes [6]. Regulatory T cells (Tregs) can release EVs carrying microRNAs (miRNAs) that interact with DCs, promoting responses such as the induction of a tolerogenic phenotype, with increased secretion of IL-10 and decreased IL-6 production following LPS stimulation [7].

In some diseases such as cancer, key functions played by EVs in the tumor microenvironment are the modification of the phenotype and function of cancer cells, the promotion of angiogenesis, and the establishment of distant pro-metastatic cell niches [8]. Brain diseases are also modulated by EV-mediated communication between neurons and glial cells, inducing the inflammation and alteration of synapses. The effects induced by brain injury include neuronal degeneration, microgliosis, and astrocytosis, which are all reduced by treatment with EVs generated by mesenchymal stromal cells [9].

Studying EVs in the context of virus infections has been crucial for demonstrating their potential contribution to viral pathogenesis since some viruses utilize EVs to counteract antiviral innate immune responses [10]. EVs generated by virus-infected cells can incorporate viral proteins and fragments of genetic material, playing a significant role in viral infection—both facilitating and suppressing it [11].

Here, we aim to provide a broad overview of the roles played by the EV-mediated delivery of miRNAs in the pathogenesis of viral infections. Despite the idiosyncrasies of many of the clinically relevant viruses, it has been recognized that the use of EVs and miRNAs is possibly evolved as a common mechanism in infection establishment. There is robust evidence in the literature suggesting that viruses use and subvert the EVs or miRNA machineries to their benefit. We aim to provide the reader with a view of the intersection between EVs and miRNAs in the context of viral diseases.

## 2. General Features of Extracellular Vesicles

EVs are constitutively released by all eukaryotic cells and allow communication in both close quarters and at a distance. Although all EVs are particles constituted by lipid layers, their cargo reflects the state of the source cell, and their content profile can be altered in adverse conditions or be manipulated by pathogens. According to their size, EVs can be classified as “small EVs” (sEVs, typically between 100 and 200 nm) and “medium/large EVs” (m/lEVs, >200 nm) [12].

The assembly of EVs is an active, energy-dependent, and regulated process [13]. Based on their biogenesis pathway, there are two distinct classes of small membrane-enclosed vesicles released from cells that differ in size and secretory mechanisms. The first of these are the "small EVs" (also referred to as exosomes by some authors), which originate from the endosome, leading to the formation of intraluminal vesicles (ILVs) by the inward budding of multivesicular bodies (MVBs). ILVs can be targeted for degradation through lysosomal pathways or can travel to the plasma membrane, where they are released into the extracellular space by the fusion of the MVB membrane with the plasma membrane [14]. The second class constitutes the “medium/large EVs”, referred to as shedding MVs, which are released into the extracellular space by the outward budding and fission of the plasma membrane [15].

The mains functions of EVs depend on their ability to interact with recipient cells and to deliver their contents. EVs originating from a variety of different cells share common structural and functional proteins, such as tetraspanins (CD9, CD63, and CD81), Rab GTPase, SNAREs, annexins, Alix, Tsg101, flotillin, cholesterol, sphingomyelin, and hexosylceramides [16]. The release of EVs is stimulated by the ATP-mediated activation of purinergic receptors [17] and activation by lipopolysaccharides [18]. EVs can use receptors for entry into target cells, and this interaction is guided by adhesion molecules such as integrins. Thus, the binding of EVs to a target cell does not occur at random but depends on the expression of specific receptors at the cell surface, many of which have not yet been identified [19].

EVs are key elements in cell-to-cell communication and have been isolated from a variety of body fluids, such as blood, semen, saliva, urine, cerebrospinal fluid, amniotic fluid, bronchoalveolar lavage fluid, synovial fluid, and breast milk, suggesting that they could be interesting biomarkers [20]. Most of their biological activity relies on the transfer of soluble and insoluble factors, such as DNA, messenger RNAs (mRNAs), microRNAs (miRNAs), proteins, lipids, and carbohydrates, reflecting the origin cell state [21]. Jeppesen et al. (2019) demonstrated that there was greater heterogeneity among extracellular vesicle populations than was previously known. For example, small EVs do not contain DNA, and smaller EVs with classical markers of exosomes, expressing CD63, CD81, and CD9, do not contain proteins such as GAPDH and HSP90 [22].

Viruses that establish long-term, persistent infections have been shown to alter the content of EVs and their biological function. On the one hand, EVs can modulate recipient cells to favor viral replication; on the other hand, EVs can restrict viral replication by triggering host immune responses. EVs require common entry pathways with viruses, using entry receptors and fusion machinery similar to those used by viruses [23]. Recently, ample evidence has been provided to show that exosomes carry and deliver viral genomes into recipient cells in vitro, as shown for hepatitis C virus (HCV) [24], hepatitis A virus (HAV) [25], and human herpesvirus 6 (HHV-6) [26]. The encapsulation of viral (RNA) genomes in host membranes may be a conserved strategy to protect against antibody-mediated neutralization. There is some evidence that human immunodeficiency virus (HIV) can utilize dendritic cell-to-T cell vesicle transfer as an alternative route for productive infection, a mechanism called transinfection, which may also be utilized by certain types of EVs [27].

miRNAs are the most studied and best characterized molecules in the class of small regulatory noncoding RNAs [28]. They are involved in several important biological processes and functions through the post-transcriptional regulation of mRNAs. Their dysregulation is often a cause or consequence of a variety of diseases such as cancer, neurodegenerative disorders, and infectious diseases [29,30,31]. Cellular miRNAs can be packaged and released in small vesicles, potentially acting as “trojan horses” by allowing genetic exchange between infected cells and uninfected neighboring ones [10,32].

Since only a select subset of the hundreds of microRNAs in a human cell end up packaged into EVs, specific mechanisms that sort microRNAs destined for export may exist. For example, Shurtleff et al. (2016) found that one RNA-binding protein, YBX1, which is a known constituent of EVs secreted from intact cells, was required for the selective packaging of miR-223 [33]. Similarly, the protein KRAS, which undergoes mutations in colorectal cancer (CRC) cells, was found to selectively increase miR-100 in EVs originating from cancerous cells. However, whether the accumulation or export of these miRNAs is a result or a consequence of oncogenic signaling remains unknown [34].

Different cells types and tissues can be characterized by their distinctive cellular miRNA landscape. An example is the human placenta, which expresses a unique set of miRNAs, mostly derived from a large cluster located on chromosome 19. Interestingly, a fraction of these placenta-enriched miRNAs is released to the extracellular environment through EVs to induce an antiviral immune response [35,36].

In viral infections, EVs carrying miRNAs can be considered useful markers of disease conditions and functional mediators of several biological processes involving cell-to-cell communication. Moreover, they have a great impact on the regulation of immune responses. Therefore, understanding the function of miRNAs in this context could help to pinpoint useful biomarkers and/or uncover potential therapeutic targets. Some details are shown in Figure 1.

## 3. DNA Viruses

### 3.1. Epstein–Barr Virus

One of the first works to suggest the link among EVs, miRNAs, and viral infection focused on Epstein–Barr virus (EBV), an enveloped, (ds)DNA, γ-herpesvirus 4 associated with infectious mononucleosis as well as with lymphoproliferative diseases [37]. It is well known that the EBV genome encodes a set of miRNAs (roughly divided into three main clusters: BHRF1, BART cluster 1, and BART cluster 2), which are essential for regulating various steps in viral pathogenesis such as latency and lytic reactivation [38]. However, it was only recently shown that EBV-infected B cells could also secrete EVs carrying viral miRNAs, which acted in a paracrine manner in other cells types, modulating their response (e.g., CXCL11 repression in DCs) [39].

EVs derived from EBV-positive lymphoblastoid cell lines (LCLs) carry, in addition to miRNAs, a diverse set of viral elements, such as the oncoprotein LMP1 and its mRNA, and host components, such as protein kinases and enzymes, which can be delivered to and regulate the activity of the target cell [40,41]. As an example mechanism of action, Yoon et al. (2016) showed that EVs from Raji cells (a Burkitt’s lymphoma cell line) could deliver miR-155 to other cell types, regulating processes such as VEGF-A production by retinal pigment epithelial cells, which could promote tumor-associated neovascularization [42].

Alternatively, EBV-infected cells can respond to host EVs to modulate viral pathogenesis. Like most herpesviruses, EBV cycles between lytic and latency phases—while in B cells the virus remains latent most of the time, the virus prefers lytic replication in epithelial cells [38] Lin et al. (2016) observed that EBV lytic reactivation in B-lymphocytes was driven by miR-200 family members, which are produced by oral epithelial cells and delivered inside EVs. According to their proposed model, the epithelial EV cargo induces B cells to reactivate the lytic cascade, allowing EBV reactivation and epithelial infection, where the virus multiplies intensely to permit its transmission through saliva [43].

Intriguingly, the miRNA cargo within EVs is not a carbon copy of the cell composition, but it shows a particular distribution pattern. Although it is suggested that miRNAs, particularly those belonging to viruses, are found in EVs because specific motifs (named EXOmotifs) are present in their sequences, not all EV-associated miRNAs are consistently present, reinforcing the involvement of other mechanisms [44].

Meckes et al. (2010) observed that EVs derived from EBV-positive nasopharyngeal carcinoma (NPC) cell lines showed a preferential enrichment of some viral miRNAs and proteins, suggesting that the virus actively interfered in the EV production machinery to favor the delivery of some elements in vesicles [45]. Considering EBV miRNA clusters, those belonging to the BART group (especially miR-BART7) are preferentially detected in NPC cell lines and their EVs, as was further validated in the plasma of patients, which is suggestive of their use as diagnostic biomarkers [46] (albeit the fact that those circulating miRNAs do not necessarily need to be shielded within EVs for stability [47]). In parallel, EV-derived LCLs were shown to contain mainly miR-BART3 and miR-BHRF1-1, which are known to modulate the inflammatory response and potentiate lytic replication, compared to the cell miRNome [48]. Therefore, the modifications induced by EBV in EV content also depend on the type of host cell that the virus infects.

In the EBV-related malignancy nasal natural killer/T-cell lymphoma (NNKTL), Komabayashi et al. (2017) highlighted the usefulness of viral miRNA carried in EVs to determine cancer stage and disease prognosis by observing a good correlation between miR-BART2-5p levels and NKKTL progression. Curiously, some circulating miRNAs were modulated in patients undergoing treatment, also pointing to their potential use as a therapeutic biomarker [49]. Similarly, Ramayanti et al. (2018) suggested that EV-associated BART13-3p detection could be a valuable way to diagnose NPC, as this method was able to distinguish this cancer from other EBV-linked cancers [50].

Considering the widespread distribution of EBV, which is estimated to infect up to 98% of the global population, it was suggested the virus could contribute to the pathogenesis of many unrelated diseases [51]. In Sjogren’s syndrome, for example, an autoimmune inflammation that promotes dysfunction of exocrine glands, it was shown that EBV-positive, B cell-derived EVs carrying miR-BART13-3p could reach salivary glands and disrupt expression of STIM1 and aquaporin 5, which resulted in compromised calcium signaling and exocrine activity; therefore, the disease phenotype was favored [52].

### 3.2. Human Papillomaviruses

Human papillomaviruses (HPV) are a large family of nonenveloped, dsDNA viruses that can be sexually transmitted and, therefore, affect the epithelial layers of the genitals, oropharyngeal, and anal mucosa. HPV infections can range from asymptomatic or benign warts to severe malignancies. Based on the virus’ oncogenic potential, they can be stratified as low- or high-risk types. Among the high-risk HPVs, HPV16 and HPV18 are the most prevalent [53].

The main HPV oncogenic proteins E6 and E7 are known to alter the host cell cycle—respectively, the tumor suppressors p53 and pRB—thus favoring its malignant transformation [54]. Not surprisingly, their activity involves alterations in the host miRNA repertoire, promoting the upregulation of protumorigenic miRNAs such as miR-21 [55]. Curiously, these onco-proteins can also modify the miRNA content in secreted EVs with the enrichment of some families, such as miR-222 [56] and the miR 17~92 cluster, which could act as tumor-promoting messengers to nearby cells [57]. The artificial expression of proteins E6 and E7 alone is enough to promote deep changes in the miRNA profile, both in the cell and the EV content, highlighting their essential role in the subversion of host mechanisms towards a favorable environment for viral establishment [58].

Among the different cancers associated with HPV, oropharyngeal squamous cell carcinoma (OPSCC) is illustrative of the hijacking abilities of the virus in EVs. In addition to being HPV^+^, OPSCC cell lines secrete a lower number of particles that are smaller in diameter. Moreover, their EVs present miRNA signatures different from their HPV-OPSCC counterparts, with a prevalence of miRNAs such as miR-99a-5p and miR-106a-5p [59]. These finding indicate that EVs carrying miRNAs could provide a more sensitive diagnostic tool for cancer classification, which could help in treatment choice.

Although the capacity of EVs to carry and deliver miRNAs is still an area of research in its infancy, many works have shown that it may be a common strategy in viral infections. Herpes simplex virus 1 (HSV-1), for example, is another member of the Herpesviridae family known to utilize miRNAs to interfere in the host cell physiology in favor of its replication and latency [60]. Similar to EBV, HSV-1 is also able to transfer viral miRNAs to neighboring cells in EVs, though its impact on infection outcome is poorly understood [61].

### 3.3. Polyomaviruses

Another example is polyomavirus JC, a dsDNA virus responsible for causing progressive multifocal leukoencephalopathy, a demyelinating disease affecting oligodendrocytes in immunocompromised settings [62]. Although the virus DNA encodes only two known miRNAs, jcv-miR-J1-5p and jcv-miR-J1-3p, these can promote viral persistence in the host (i) by reducing the expression of the protein large T antigen (LTAg), favoring a state of latency instead of active replication, and (ii) by downmodulating the host protein ULBP3 (UL16-binding protein 3), a ligand for the receptor NKG2D (natural killer group 2, member D) that is required for NK cell-mediated killing [63]. Considering that both miRNAs were efficiently detected within and transferred by EVs produced by the infected hematopoietic cell line KG-1 [64], it has been suggested that they can prepare neighboring cells for future infection.

Also belonging to the Polyomaviridae family and closely related to the JC virus, the BK virus can promote nephropathy and is frequently linked to kidney graft failure [65]. Kim et al. (2017) demonstrated that urinary EVs were important sites for BK virus-derived miRNA accumulation. Furthermore, those miRNA levels showed a better correlation to viral burden than the conventional search for viral DNA in plasma or urine, suggesting that it would be a better biomarker for disease monitoring [66].

Following this rationale, van der Ree et al. (2017) showed that circulating miRNAs could also be an interesting biomarker in hepatitis B virus (HBV) infection. Although miRNAs associated with antigenic status and treatment response can circulate bound to hepatitis B surface antigen (HBsAg) particles, some miRNAs such as miR-192-5p, miR-193b-3p, and miR-194-5p could be detected within EVs, and their levels were considered to be good disease-monitoring parameters [67]. From an immunological point of view, Enomoto et al. (2017) highlighted the function of miR-192 and four other exosomal miRNAs (miR-21, miR-215, miR-221, and miR-222) in repressing IL-21 expression, a T cell-derived cytokine with anti-HBV activity [68].

## 4. RNA Viruses

EVs have been recognized as an important player in the pathogenesis of RNA virus infections, involved in the delivery of viral and host components that contribute to disease establishment, but also employed as a communication strategy of the host defense to restrict viral spread to uninfected cells [69].

### 4.1. Respiratory Viruses

In infection by rhinovirus, for example, Gutierrez et al. (2016) observed increased levels of miR-155 loaded in EVs from nasal secretions, which could be a way to promote the host airway immunity because this miRNA could induce inflammatory genes [70]. For another respiratory RNA virus, however, the delivery of miRNAs by serum EVs could be detrimental to the host response. Okamoto et al. (2018) showed that miR-451a was normally found in circulating EVs, and that it could downmodulate the response of macrophages and dendritic cells to inactivated influenza A virus (IAV), reducing the expression of IL-6 and IFN-β, and thus potentially affecting the efficiency of vaccination or even the outcome of natural infection [71]. Recently, Scheller et al. (2019) uncovered miR-17-5p as another EV-associated miRNA that helped IAV infection by modulating the expression of antiviral factor Mx1 [72]. Thus, IAV may substantially subvert the host miRNA repertoire to its own benefit.

### 4.2. HIV

Human immunodeficiency virus (HIV) is one the most studied viruses in the world, and the involvement of EVs in the pathogenesis of HIV infection is well-known [73]. However, the relevance of miRNAs in this context is still being uncovered. For example, Roth et al. (2016) reported that HIV-infected macrophages showed enrichment of many miRNAs in secreted EVs, particularly miR-29a, miR-150, miR-518f, and miR-875 [74], but the individual roles of these miRNAs in HIV pathogenesis requires further characterization. Many papers have drawn hypotheses from databases for target prediction, but the functions of only a limited number of the identified miRNAs have been comprehensively validated. Thus, a great area for experimental validation is open for future research. Still, some key roles played by EV-delivery miRNAs in HIV infection have already been described in the literature.

Hubert et al. (2015) observed that HIV-1^+^ patients (naïve of antiretroviral therapy) showed an increased abundance and size of their plasma EVs content, and these parameters were also associated with disease progression [75]. Within these EVs, the authors also noted an increased accumulation of miR-155 and miR-223 in those subjects [75]. Intriguingly, Liao et al. (2017) reported that the depletion of serum EVs in culture media enhanced the replication of HIV in vitro by altering the expression of HIV co-receptors and the metabolic activity of the cells, but no influence from miRNAs was detected [76]. Whether these findings can be reproduced in vivo remains unknown.

EVs from HIV-infected cells carry viral proteins, such as Nef [77]. Because of its ability to modulate the endosomal trafficking network, Nef can actively interfere in the biogenesis of EVs, altering their composition and release [78]. Not surprisingly, Nef^+^ EVs also show a particular miRNA signature, with reduced levels of miRNAs such as miR-16, miR-125b, and miR-146a, which are known to promote immune response and restrict infection [79].

Interestingly, the authors showed in another work that some of these miRNAs increased in Nef^+^ cells, and their target mRNAs were enriched in EVs derived from these cells [80]. Additionally, Nef interacts with the protein Argonaute-2, compromising the activity of the RNA-induced silencing complex (RISC) and, therefore, affecting the splicing activity driven by miRNA/mRNA binding [81]. Hence, it is possible that HIV creates a network in which selected miRNAs are enriched in EVs for delivery to neighboring cells. In concert with viral proteins such as Nef, EVs alter the expression of some genes, and their products are further exported, amplifying viral effects in the host.

Chronic immune activation is another hallmark in HIV infection that can lead to a compromised neurological function from a large set of mechanisms [82]. Yang et al. (2018) observed that miR-9 was considerably enriched in EVs from astrocytes stimulated with Tat protein. miR-9 has an important function in recruiting and activating microglia by downmodulating the protein PTEN, thus contributing to immune activation in neuronal tissue [83]. In another work by Yelamanchili et al. (2015), the authors highlighted the role of EVs carrying miR-21 in promoting neuronal death. In a model of SIV (simian immunodeficiency virus) encephalitis, they reported an increased production of this miRNA in the brain tissues of infected animals. Derived from microglial cells, encapsulated miR-21 could promote necroptosis in neurons in a TLR7-dependent way [84], pinpointing another mechanism of viral neuropathology.

HIV infection is commonly associated with risk factors such as drug abuse, particularly injected drugs such as opioids. Using the SIV model, Hu et al. (2012) showed a synergic effect between SIV infection and morphine administration in compromising the viability of neuronal cells, which could contribute to the neurocognitive deficits in these subjects [85]. Interestingly, this neuronal death occurs through the release by astrocytes of EVs carrying miR-29b, which act by repressing the expression of the neurotropic factor platelet-derived growth factor B [85]. In another example of HIV–drug interaction, Sharma at al. (2018) observed an increased production of EVs in a model of human macrophages infected with HIV and exposed to cocaine [86]. Interestingly, these vesicles were free of viral proteins and could promote pulmonary arterial smooth muscle cell proliferation from the delivery of miR-130a, which activates the PI3K/Akt pathway, and could be a mechanism explaining the pulmonary arterial hypertension observed in HIV patients [87].

HIV not only subverts the host miRNA repertoire to its benefit, but it also codes its own set of miRNAs in its genome, mostly belonging to the TAR (transactivation response elements) group [88]. Narayanan et al. (2013) showed that EVs from HIV-infected T cell lines carried TAR miRNAs mostly in their precursor form, but also the RNases Dicer and Drosha, suggesting that TAR miRNAs could mature once they were delivered to a naïve cell [89]. Interestingly, TAR miRNAs counterbalance apoptosis in the targeted cell by decreasing the expression of the pro-apoptotic protein Bim, favoring their infection by the virus [90]. Additionally, TAR miRNAs could induce the production of inflammatory cytokines such as IL-6 and TNF-β by activating TLR3, contributing to HIV-induced inflammation [91].

### 4.3. Hepatitis C Virus

Hepatitis C virus (HCV) is another RNA virus of public health concern. It is a flavivirus that is considered an important cause of liver disease worldwide. In addition, it known to utilize EVs as a transmission vessel in order to evade the host immune response [31]. EVs produced by infected hepatoma cells carry viral RNA and proteins, and they can infect neighboring cells without the involvement of HCV entry receptors or even in the presence of neutralizing antibodies [92,93]. In addition, these infectious EVs carry host components, such as miR-122 and the proteins Argonaute-2 and HPS90, which help to stabilize the viral RNA and enhance its infectivity [93].

Mechanistically, this delivery of an altered content of host factors affects the activity of vesicle-targeted cells. For example, EVs derived from HCV-infected hepatocytes can reach hepatic stellate cells, inducing a profibrogenic profile. This effect is mediated by miR-19a, which promotes the STAT3/TGF-β1/Smad3 signaling pathway and cell division [94]. This altered activity of stellate cells is also reflected in their miRNA production, and some EV-associated miRNAs detected in the serum of HCV patients are believed to mirror their dysfunction [95]. As liver fibrosis precedes the development of cirrhosis and hepatocellular carcinoma (HCC), miR-19a could be an interesting target for the treatment of HCV infections.

Even though cancer staging can be a valuable tool in disease and treatment monitoring, there is no consensus yet on the best system for HCC staging [96]. Still, EV-associated miRNAs could become an effective biomarker for this purpose. Liu et al. (2015) showed the prominent use of miR-122 as a way to monitor HCC development. Although their findings were observed in a rat model, requiring further validation in human samples, their method showed a superior response compared to the current dosage of alpha-fetoprotein in blood [97]. Curiously, high levels of circulating miR-122 were also considered to be a predictive factor for successful antiviral therapy response among patients infected with different genotypes of HCV [98], though its levels tended to be higher in patients with genotype 1b [99].

Regarding therapeutic approaches to treat hepatitis C, the use of direct-acting antiviral drugs (DAA), in replacement of IFN-based therapy, was shown to not only interfere directly with viral replication but also promote host immunity. Santangelo et al. (2018) observed that EVs from chronic HCV patients presented a particular miRNA signature and were able to downmodulate NK cell activity. However, after DAA therapy, their EVs recovered the healthy miRNA profile and could no longer inhibit NK cell function [100].

The use of immune stimulatory agonists was also investigated as coadjutants in therapy. The TLR3 ligand PolyIC, for example, can activate macrophages and induce the production of EVs enriched in miR-29. These vesicles can then act on hepatocytes and activate IFN-induced genes, promoting an effective antiviral state and protecting cells from infection [101]. In a similar rationale, Qian et al. (2016) observed the strong anti-HCV activity of EVs derived from umbilical cord mesenchymal stem cells [102]. Interestingly, these cells constitutively secreted EVs harboring miRNAs that could repress HCV replication in vitro (let-7f, miR-145, miR-199a, and miR-221). Considering that the administration of these EVs showed a synergic effect with the anti-HCV drugs IFN-α and telaprevir, EVs could be interesting therapeutic coadjutants in HCV treatment.

### 4.4. Japanese Encephalitis Virus

Japanese encephalitis virus (JEV) is another flavivirus. JEV is recognized as an important cause of viral encephalitis, and neuroinflammation is a hallmark of the infection [103]. JEV was shown to modulate the miRNA expression profile in infected microglial cells, favoring responses associated with enhanced cell cycling and suppression of the endothelial barrier and immune response, thus contributing to the disease phenotype [104]. However, JEV was also shown to induce the production of EVs loaded with miR-let-7a/b (but free of viral RNA and proteins) by infected microglia, which could be delivered to bystander cells. There, those miRNAs induced the activation of apoptotic caspases, which could promote neuronal damage [105]. Thus, JEV-associated neurotoxicity does not necessarily require the presence of viral components. It will be interesting to evaluate the relevance of EV-associated miRNAs in the pathogenesis of other flaviviruses, especially Zika virus, our understanding of which has only been developed in recent years.

Instead of being a particular mechanism developed by some viruses to promote disease, the usage of miRNA-loaded EVs sounds most likely to be the rule in the pathogenesis of viral infections. Recent work by Germano et al. (2019) observed that Coxsackievirus B, an important cause of infectious myocarditis, promoted the release of miR-590-5p within EVs by cardiomyocytes. Furthermore, this miRNA could enhance viral proliferation by modulating the protein sprouty-1, a tumor suppressor that controls the cell cycle [106]. Thus, new lines of investigation should uncover how these mechanisms operate for other viruses in the future. We summarize the main effects of miRNAs on viral infections in Figure 2.

## 5. Conclusions and Future Perspectives

The study of EVs carrying miRNAs in the context of viral infections is important for the understanding of fundamental disease mechanisms. EVs have emerged as specific carriers of cellular and viral components, including miRNAs, proteins, and viral genomes, and they can be produced during both active viral replication and during viral latency. Future studies that attempt to unravel the structure and cargo of EVs produced by infected cells, along with the fine mechanisms by which they affect viral infection, is a key step for the future use of EVs as tools for the diagnosis of disease and/or the exploitation of new drug targets.

## Figures and Tables

**Figure 1 cells-08-00611-f001:**
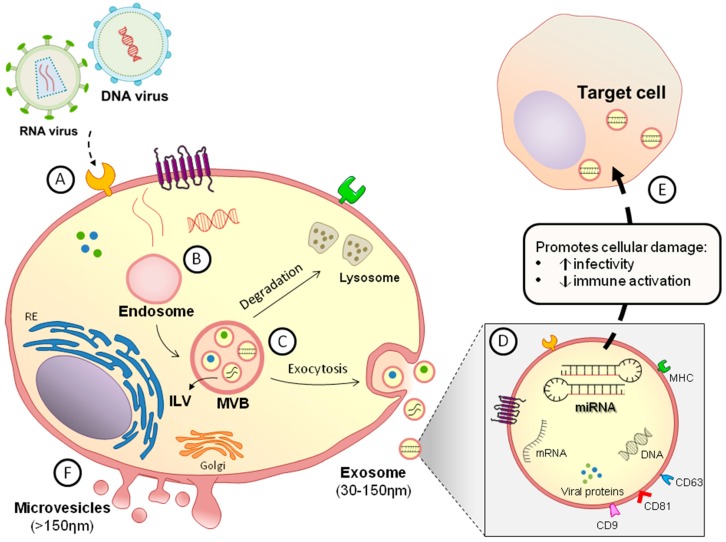
Biogenesis of extracellular vesicles (EVs) carrying micro RNAs (miRNAs) in viral infection. (A) After infection of the cells, different viral components (proteins and genetic material) are available in the cytoplasm and (B) can be encapsulated by endosomes generating endocytic vesicles. (C) During the biogenesis of small EVs (exosomes), formation of intraluminal vesicles (ILVs) occurs by the inward budding of endosomal multivesicular bodies (MVBs), which can be delivered to and degraded by the lysosomal pathway or follow an exocytic pathway that leads to the plasma membrane, where they fuse and release the EVs from the cell. (D) EVs can carry functional biomolecules, which may be of viral origin or be from the infected cell itself, such as proteins, DNA, messenger RNAs (mRNAs), and miRNAs, which have important immunomodulatory effects. In addition, EVs carry a profile of membrane components similar to those of the source cell in addition to tetraspanins (CD63, CD81, and CD9). (E) Infection can stimulate the synthesis of different families and types of miRNAs capable of acting on the target cell and contribute to the establishment of the infection, increasing its infectivity and/or diminishing immune activation. (F) The release of medium EVs (microvesicles) occurs via a pathway independent of the formation of MVBs and exocytosis, shedding directly from the cytoplasmic membrane.

**Figure 2 cells-08-00611-f002:**
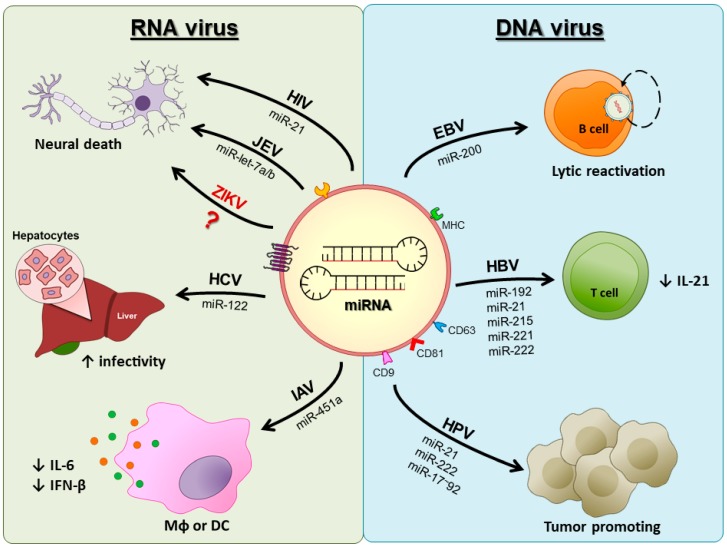
The modulating effect of miRNAs on different viral infections. Several RNA/DNA viruses have been described to promote the secretion of EVs carrying different miRNAs that promote modulation in the target cell and contribute to the establishment of infection. Note: HIV, human immunodeficiency virus; JEV, Japanese encephalitis virus; ZIKV, Zika virus; HCV, hepatitis C virus; IAV, influenza A virus; EBV, Ebola virus; HBV, hepatitis B virus; and HPV, human papilloma virus.

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
