# Peer review of "Delivery of microRNAs by Extracellular Vesicles in Viral Infections: Could the News be Packaged?"

_cells, 2019, doi:10.3390/cells8060611_

Round 1
Reviewer 1 Report
Yoshikawa et al review the role of extracellular vesicles and their contents in infection with RNA and DNA virus. This in an interesting review article summarizing the role of these vesicles in cells. Whereas the content is mostly fine, there are many typos and unclear sentences that need attention, especially in the abstract, sections 1 and 2.
Major comments
There is some confusion in the beginning of the article about the terms EVs, exosomes, MVB, ILV, etc. What are the characteristics of these vesicles and what the authors consider that it all includes. Possibly this should be clarified with a table showing the main features of these vesicles.
Some of the features of EVs are in Figure 1, but not all are labelled, such as the endosomes and the MVs.
The sections 3 and 4 about DNA and RNA virus should include subtitles for each virus (e.g. for EBV, HPV, Polyoma, HBV, HIV, HCV, JEV, Flavivirus, Zika, Coxdackievirus B, etc).
Some of the text in the figures is rather small, the font should be increased (e.g. the name of the receptors, RNA virus, DNA virus in Figure 1, or MO on DC in Figure 2).
Line 192: E6 is altering p53, E7 is altering pRB
Line 190: All 14 high-risk HPV types are linked to cervical cancer (and some other cancers), HPV16 and HPV18 are to most prevalent ones
The manuscript should be proofread by a native English-speaking scientist since there are many typos and unclear, grammatically incorrect sentences throughout the manuscript (some are listed below).
Minor comments
Line 12: cells
Line 20: last sentence not clear
Line 28: were considered “cell dust”
Line 28: has great importance
Line 38: in order to study
Line 59: utilize
Line 116: are depicted in
Line 121: to also be …not clear
Line 122: initially the formation… not clear
Line 139: could also secrete
Line 163: to favor the
Line 166: in the plasma
Line 166: laboratorial… diagnostic
Line 218: linked to
Line 269: it is possible that
Line 270: a network in which
Line 270: enriched in EVs for delivery
Line 271: as Nef, EVs alter
Line 283: risk factors such as
Line 288: In another example
Line 318: as a way to monitor
Line 325: interfere directly with the
Line 330 and 338: coadjutants…. Not clear
Line 349: only in recent
Line 360: to promote secretion of EVs
Author Response
We thank the editorial office and the referees for reading our manuscript and providing suggestions to our work. We address below our replies for the considerations arose. The edits in the manuscript are highlighted in yellow.
“Yoshikawa et al review the role of extracellular vesicles and their contents in infection with RNA and DNA virus. This in an interesting review article summarizing the role of these vesicles in cells. Whereas the content is mostly fine, there are many typos and unclear sentences that need attention, especially in the abstract, sections 1 and 2.”
COMMENT: There is some confusion in the beginning of the article about the terms EVs, exosomes, MVB, ILV, etc. What are the characteristics of these vesicles and what the authors consider that it all includes. Possibly this should be clarified with a table showing the main features of these vesicles.
RESPONSE: We apologize and agree that some information was confusing. Some sentences have been changed to make the terms used in the manuscript clearer in sections 1 and 2.
COMMENT: Some of the features of EVs are in Figure 1, but not all are labelled, such as the endosomes and the MVs.
RESPONSE: We did changes in Figure 1, such as the inclusion of the endosome and the indication of the ILVs, and increased the font size. We have also made changes to the legend to make it clearer.
COMMENT: The sections 3 and 4 about DNA and RNA virus should include subtitles for each virus (e.g. for EBV, HPV, Polyoma, HBV, HIV, HCV, JEV, Flavivirus, Zika, Coxdackievirus B, etc).
RESPONSE: We thank the referee for the suggestion; we have added subsections to the manuscript.
COMMENT: Some of the text in the figures is rather small, the font should be increased (e.g. the name of the receptors, RNA virus, DNA virus in Figure 1, or MO on DC in Figure 2).
RESPONSE: We thank the referee for the suggestion; we have increased the font size in Figures 1 and 2.
COMMENTS: Line 192: E6 is altering p53, E7 is altering pRB AND Line 190: All 14 high-risk HPV types are linked to cervical cancer (and some other cancers), HPV16 and HPV18 are to most prevalent ones
RESPONSE: We have corrected this information in the manuscript
COMMENTS: The manuscript should be proofread by a native English-speaking scientist since there are many typos and unclear, grammatically incorrect sentences throughout the manuscript (some are listed below).
Minor comments * Line 12: cells / Line 20: last sentence not clear / Line 28: were considered “cell dust” / Line 28: has great importance / Line 38: in order to study
Line 59: utilize / Line 116: are depicted in / Line 121: to also be …not clear / Line 122: initially the formation… not clear / Line 139: could also secrete / Line 163: to favor the / Line 166: in the plasma / Line 166: laboratorial… diagnostic / Line 218: linked to / Line 269: it is possible that / Line 270: a network in which / Line 270: enriched in EVs for delivery / Line 271: as Nef, EVs alter / Line 283: risk factors such as / Line 288: In another example / Line 318: as a way to monitor / Line 325: interfere directly with the / Line 330 and 338: coadjutants…. Not clear / Line 349: only in recent / Line 360: to promote secretion of EVs
RESPONSE: We apologize for the typos and grammar mistakes. The manuscript has been proofread by the MDPI English Editing service.
Reviewer 2 Report
The paper by Yoshikawa et al is a review paper looking into the role played by exosomes during viral infections. The main focus is the role of miRNA cargo in exosomes during virus infection. A major concern is that the language and style of writing are very poor. In certain instances, it was hard to make sense of the writing and it is recommended that the authors use and English writing/editing service to improve the quality of the language. Furthermore, relevant recent publications are missing in the text and citation list. Examples include: Dongen et al. 2016 (https://mmbr.asm.org/content/80/2/369); Jeppesen et al., 2019 (https://www.cell.com/cell/pdf/S0092-8674(19)30212-0.pdf). A second concern is that the review is mostly stating observations, but provides minimal critique or meaning of results in cited publications. For example, what are the names of the polyomavirus miRNAs? How do they specifically regulate viral persistence? What are the specific targets of the miRNAs? The authors must also include the mechanism(s) by which miRNAs are sorted into exosomes (e.g. see Shurtleff et al https://elifesciences.org/articles/19276); Cha et al (https://elifesciences.org/articles/07197) etc. Since the paper was focused on miRNA cargo in exosomes, more details are required. For HCC (line 317-323), the authors must include a statement that HCC statging is currently not universally adopted hence the potential need to use the exosomes for staging the disease. These issues are major and the paper needs major revisions to be reconsidered for publication.
Author Response
We thank the editorial office and the referees for reading our manuscript and providing suggestions to our work. We address below our replies for the considerations arose. The edits in the manuscript are highlighted in yellow.
The paper by Yoshikawa et al is a review paper looking into the role played by exosomes during viral infections. The main focus is the role of miRNA cargo in exosomes during virus infection.”
COMMENT: A major concern is that the language and style of writing are very poor. In certain instances, it was hard to make sense of the writing and it is recommended that the authors use and English writing/editing service to improve the quality of the language.
RESPONSE: We apologize for the typos and grammar mistakes. The manuscript has been proofread by the MDPI English Editing service.
COMMENT: Furthermore, relevant recent publications are missing in the text and citation list. Examples include: Dongen et al. 2016 (https://mmbr.asm.org/content/80/2/369); Jeppesen et al., 2019 (https://www.cell.com/cell/pdf/S0092-8674(19)30212-0.pdf).
RESPONSE: We thank the referee for the suggestion. Some information was added regarding these works in section 2 of this manuscript.
COMMENT: A second concern is that the review is mostly stating observations, but provides minimal critique or meaning of results in cited publications. For example, what are the names of the polyomavirus miRNAs? How do they specifically regulate viral persistence? What are the specific targets of the miRNAs?
RESPONSE: We apologize for the lack of clarity. The two miRNAs encoded by polyomavirus JC (jcv-miR-J1-5p and jcv-miR-J1-3p) can promote viral persistence by two mechanisms: reduction of the expression of the viral protein LTAg, favoring latency instead of viral replication, and downregulation of the host protein ULBP3, the ligand for the NK cell activator receptor NKG2D. We added those considerations in the manuscript.
COMMENT: The authors must also include the mechanism(s) by which miRNAs are sorted into exosomes (e.g. see Shurtleff et al https://elifesciences.org/articles/19276); Cha et al (https://elifesciences.org/articles/07197) etc. Since the paper was focused on miRNA cargo in exosomes, more details are required.
RESPONSE: We thank the referee for the suggestion. Some details about miRNA cargo demonstrated in these works were added in section 2.
COMMENT: For HCC (line 317-323), the authors must include a statement that HCC statging is currently not universally adopted hence the potential need to use the exosomes for staging the disease. These issues are major and the paper needs major revisions to be reconsidered for publication.
RESPONSE: We have added the statement in the manuscript.
Reviewer 3 Report
This review article by Yoshikawa and others broadly summarizes relationships between virus infection and EVs, and EVs’ effects on the infectious diseases and even usefulness in diagnosis. Specific explanation to each virus is very limited, but this may be unavoidable because of the nature of a broad area review.
Writing is grammatically incorrect in some parts (eg, first paragraph of Introduction, line 139, line 146, 163) and must be checked by native English speakers, overall. Some sentences are overstuffed and thus hard to follow (eg, the first sentence of Introduction).
Author Response
We thank the editorial office and the referees for reading our manuscript and providing suggestions to our work. We address below our replies for the considerations arose. The edits in the manuscript are highlighted in yellow.
“This review article by Yoshikawa and others broadly summarizes relationships between virus infection and EVs, and EVs’ effects on the infectious diseases and even usefulness in diagnosis. Specific explanation to each virus is very limited, but this may be unavoidable because of the nature of a broad area review.”
COMMENT: Writing is grammatically incorrect in some parts (eg, first paragraph of Introduction, line 139, line 146, 163) and must be checked by native English speakers, overall. Some sentences are overstuffed and thus hard to follow (eg, the first sentence of Introduction).
RESPONSE: We apologize for the typos and grammar mistakes. The manuscript has been proofread by the MDPI English Editing service.
Round 2
Reviewer 1 Report
The manuscript is much improved, all the suggestions and corrections were sufficiently addressed in the revised version of the manuscript. Some minor corrections.
Some additional corrections
Line 25: 1. Introduction
Line 67: and miRNA possibly evolved
Line 117: and function
Line 121: allowing genetic exchange between cells, and in particular between infected..
Line 325: [49] reference needs formatting
Line 342: it is known
Line 353: are believed
Line 361: current measurement
Line 378: a synergistic effect
Lien 383: that is a hallmark
Reviewer 2 Report
Yoshikawa et al revised their paper reviewing the role of miRNAs in exosomes from virus-infected cells. Exososmes are increasingly becoming important in the biology of virus infections and other human pathologies. Following the revision, it is recommended to accept the paper for publication.
Line 25: please correct the section numbering